# Hair Allopregnanolone in Mares and Foals as a Retrospective Biomarker of Predicting Feto-Maternal Well-Being

**DOI:** 10.3390/ani15060768

**Published:** 2025-03-07

**Authors:** Nicola Ellero, Aliai Lanci, Jole Mariella, Robin van den Boom, Alessio Cotticelli, Tanja Peric, Alberto Prandi, Francesca Freccero, Carolina Castagnetti

**Affiliations:** 1Department of Veterinary Medical Sciences (DIMEVET), University of Bologna, Via Tolara di Sopra 50, Ozzano dell’Emilia, 40064 Bologna, Italy; nicola.ellero3@unibo.it (N.E.); aliai.lanci2@unibo.it (A.L.); jole.mariella2@unibo.it (J.M.); francesca.freccero2@unibo.it (F.F.); carolina.castagnetti@unibo.it (C.C.); 2Department of Clinical Sciences, Faculty of Veterinary Medicine, University of Utrecht, Yalelaan 114, 3584 CM Utrecht, The Netherlands; r.vandenboom@uu.nl; 3Department of Veterinary Medicine and Animal Production, Federico II University, via Delpino 1, 80137 Naples, Italy; 4Department of Agricoltural Food, Environmental and Animal Science (DI4A), University of Udine, Via delle Scienze 206, 33100 Udine, Italy; tanja.peric@uniud.it (T.P.); alberto.prandi@uniud.it (A.P.); 5Health Science and Technologies Interdepartmental Center for Industrial Research (CIRI-SDV), University of Bologna, Via Tolara di Sopra 50, Ozzano dell’Emilia, 40064 Bologna, Italy

**Keywords:** neonatal foal, mare, pregnancy, prenatal, hair, hormones, pregnanes, allostasis, biomarker

## Abstract

Successful pregnancies require several adaptations in the mare, and steroid hormones induce, organize, and maintain many of these adaptations. Allopregnanolone (ALLO), a neurosteroid and progesterone metabolite, is of particular interest. During pregnancy, ALLO is produced in increasing amounts by the mare, the placenta, and the fetal brain, gonads, and adrenal glands. Since a single hair sample taken at birth offers insights into a prolonged yet identifiable prenatal period, the authors examined the possible role of ALLO as a biomarker predicting feto-maternal well-being by the hair of both mares with normal or high-risk pregnancies and their respective healthy and sick foals. Since the prenatal activity of the hypothalamus–pituitary–adrenal axis is crucial for the ultimate maturation of the fetus and its adaptation to extra-uterine life, the lower ALLO concentrations in sick foals observed in the present study deserves further attention as a potential new biomarker of prenatal disease.

## 1. Introduction

Assays of steroid hormone concentrations in hair offer a promising alternative for studies focused on the end of pregnancy. A single hair sample taken at birth offers insights into a prolonged yet identifiable prenatal period and avoids invasive blood collection from the fetus. Equine fetal hair begins to grow around day 270 of pregnancy [1,2]. This suggests that the hormone concentrations measured in hair samples taken at birth mirror their accumulation over the last two months of pregnancy. The hair grows 1 cm per month in humans [3] and at a similar rate in foals [4]. As hair is always shaved close to the skin rather than being plucked, the portion of hair located beneath the scalp is not collected and the last 15 days of pregnancy are not represented in the sample.

In the equine species, the assessment of prenatal hypothalamus–pituitary–adrenal (HPA) axis activity using this novel approach has been performed in recent studies [4,5,6,7]. Among the main findings, increased hair dehydroepiandrosterone-sulfate (DHEA-S) concentrations and a decreased cortisol/DHEA-S ratio may serve as potential biomarkers for high-risk pregnancy, along with markers of resilience and allostatic load in sick foals.

The metabolism of pregnanes in late gestation is intricate, involving the fetus, chorioallantois, and endometrium. ALLO, a neuroactive progesterone metabolite, is known to rise throughout pregnancy, reaching its peak during late gestation in mares [8,9]. It is known that during this period, the mare produces a wide number of pregnanes, including ALLO, which in its bioactive form has important functions in maintaining pregnancy [8]. ALLO plays a crucial role in signaling pregnancy to the mare’s brain, triggering opioid production in the brainstem, and amplifying GABAergic activity within the paraventricular nucleus. This process helps modulate the HPA axis response to stress during late pregnancy [10]. Such an adaptation is thought to shield the fetus from the detrimental effects of maternal glucocorticoids, which could otherwise contribute to adverse early life programming [10].

Rossdale et al. [11] examined alterations in circulating pregnanes in mares during late gestation, focusing on the presence of several clinical abnormalities such as placentitis, placental edema, placenta hypoplasia, and twin pregnancies. As described by Legacki et al. [9], significant declines in physiological ALLO concentrations begin from 5 days before parturition. In the study performed by Wynn et al. [12], increased ALLO concentrations were found in the plasma of mares with chronic placentitis. These studies indicate that increased pregnanes at term are associated with placental dysfunction and fetal stress.

In equine neonates, the experimental intravenous administration of ALLO in healthy newborn foals resulted in alterations in neurological behavior and consciousness, resembling those observed in foals affected by neonatal maladjustment syndrome. This supports the crucial role of ALLO in neuronal function [13]. Increased ALLO concentrations have also been documented in critically ill equine neonates [14].

The present study aimed to evaluate ALLO concentrations in the hair of mares and foals in relation to the selected mare’s clinical parameters, including age, parity, gestation length, and type of pregnancy (normal vs. high-risk); the selected foal’s clinical parameters, including sex, weight, Apgar score, outcome, clinical condition (healthy vs. sick); and the different diagnoses of sick foals (prematurity vs. dysmaturity vs. hypoxic-ischemic encephalopathy). An additional objective was to investigate the predictive value of ALLO concentrations in the clinical outcome of mares and foals. The study is based on the hypothesis that (i) ALLO concentrations measured in foal’s hair are influenced by those of the respective mare and (ii) sick foals have different ALLO concentrations compared to healthy ones.

## 2. Materials and Methods

### 2.1. Population

Mares and their foals included in the study were either hospitalized for attended parturition or admitted less than 24 h after foaling due to foal’s clinical condition at the Veterinary Teaching Hospital (VTH) of the University of Bologna, Italy. The population of mare–foal pairs was included in the study according to a convenience criterion, as it represents the population afferent to the VTH in the 2024 breeding season, and categorized into healthy (group H) and sick (group S) foals based on history and complete clinical assessment conducted at birth or admission.

Mares were hospitalized around day 310 of pregnancy for monitored parturition. The animals were kept in individual, spacious boxes with straw bedding, provided with hay ad libitum and concentrates twice daily, and allowed free grazing throughout the day. Upon admission, a comprehensive clinical assessment was conducted, which included hematology, biochemistry, and transrectal ultrasonography for pregnancy monitoring. Transabdominal ultrasonography was performed as needed, based on the clinicians’ assessment. Mares were then monitored clinically twice daily and underwent ultrasonographic evaluations every 5 to 10 days until parturition. If an increase in the combined thickness of the uterus and placenta (CTUP; reference range <12 mm, [15]) and/or vulvar discharge were observed, a cervical swab was taken for bacterial culture. After delivery, a macroscopic examination of the fetal membranes was conducted in all mares, with samples being collected for histopathological analysis in those with high-risk pregnancies.

A high-risk pregnancy was characterized by a history of premature udder development or lactation, increased CTUP, serosanguineous or purulent vulvar discharge, systemic illness in the mare, or abnormalities identified through gross and histopathological examination of the fetal membranes. Mares diagnosed with high-risk pregnancies were managed based on the clinicians’ discretion.

Upon birth, all foals delivered at the VTH underwent a comprehensive clinical evaluation, which included hematology and biochemistry. Blood cultures and arterial blood gas analyses were performed at the discretion of the clinicians when deemed necessary. In some cases, mare–foal pairs were admitted to the VTH within 24 h after foaling due to the foal’s clinical condition. All sick foals received a thorough clinical assessment, encompassing blood culture, hematology, biochemistry, arterial blood gas analysis, and serum IgG measurement. Both mares and foals were continuously monitored during the hospitalization, with clinical examinations conducted every 2 to 12 h, depending on the severity of their condition.

Foals delivered from normal pregnancies were classified as healthy if they achieved an Apgar score [16] ≥8 and exhibited normal clinical findings during hospitalization, which included a serum IgG concentration >800 mg/dL at 12–24 h of life.

Foals born at the VTH or referred after birth were classified as sick only if they presented with conditions linked to intra uterine life. Diagnosis of hypoxic-ischemic encephalopathy was made when a history of hypoxic insult was confirmed, accompanied by clinical signs, particularly those suggestive of neurological dysfunction, and after ruling out other neurological disorders such as meningitis or trauma. Key historical factors typically included high-risk pregnancies, while common clinical signs encompassed the loss or absence of the suckling reflex, abnormal teat-seeking behavior, dysphagia, hyperreactivity, and weakness. Foals diagnosed with hypoxic-ischemic encephalopathy, where dystocic parturition was evident, were excluded. Sick foals were diagnosed with prematurity when born prior to day 320 of gestation and dysmaturity when born after day 320 but with immature physical characteristics (e.g., low body weight and inability to maintain body homeostasis).

Data recorded for mares included breed, age (years), parity, type of pregnancy (normal/high-risk), prepartum treatment (yes/no), and gestation length (days). Data recorded for foals included breed, sex, weight (kg), Apgar score in foals born at the VTH, age at admission in foals admitted after birth (hours), diagnosis, and outcome.

### 2.2. Sample Collection

Hair samples were obtained using a Heiniger^®^ clipper (Industrieweg 8–3360, Heiniger, Herzogenbuchsee, Switzerland) from the sternal area in foals and the dorsal cervical region in mares within 24 h of birth, admission, or parturition, by shaving the hair close to the skin. The samples were then stored in paper envelopes, kept in the dark at room temperature, and sent to the Laboratory of Veterinary Physiology and Reproductive Pathophysiology at the University of Udine (Department of Agricultural, Food, Environmental, and Animal Science), where the analyses were carried out.

### 2.3. Hormone Analysis

Hair samples were processed for steroid analysis and examined following the procedure outlined by Peric et al. [17], who recently validated the ELISA technique for equine hair. Briefly, 200 mg of hair was rinsed with 5 mL of ultrapure water, followed by a wash with 5 mL of isopropanol to remove any surface steroids. Steroids were extracted from the hair using 3 mL of methanol. ALLO concentrations were measured using a commercial ELISA kit (DetectX Allopregnanolone kit; Arbor Assays^®^, 1143 Highland Dr., Ann Arbor, MI, USA). The assay exhibited a sensitivity of 50.4 pg/mL, with intra- and inter-assay coefficients of variation of 6.4% and 11.0%, respectively.

### 2.4. Statistical Analysis

The Shapiro–Wilk test was applied to assess the data distribution. As the data were not normally distributed, non-parametric tests were utilized. To accurately study the relative abundance of the neurosteroid within the mare–foal pairs, a proper approach was necessary, and the ratio between the foal and mare ALLO concentrations was calculated. The Spearman’s rank correlation coefficient was used to correlate foals’ and mares’ hair ALLO concentrations within the same group and, in addition to their ratio, with mare’s age, parity, gestation length, and foals’ weight and Apgar score.

The Mann–Whitney test was applied to assess differences in ALLO concentrations in foals based on sex, outcome (survivors vs. non-survivors), and maternal ALLO concentrations in relation to fetal sex; variations in ALLO concentrations and their ratio in mares and foals between the H (healthy) and S (sick) groups, as well as between sick foals born at the VTH and those admitted post-birth; and differences in gestation length, foal weight, and Apgar scores between the H and S groups.

The Kruskal–Wallis test was applied based on data distribution, followed by post hoc analysis, to test differences in mares’ and foals’ ALLO concentrations and their ratio between healthy and sick foals classified according to diagnosis (i.e., prematurity, S-PRE, sub-group vs. dysmaturity, S-DYSM, sub-group vs. hypoxic-ischemic encephalopathy, S-HIE, sub-group); differences in foal’s weight among H group, S-PRE, S-DYSM, and S-HIE; and differences in mare’s gestation length among H group, S-PRE, S-DYSM, and S-HIE.

Finally, ALLO concentrations were log-transformed and included in a logistic regression model to study whether hormone concentrations (as an independent variable) could predict the clinical outcome (as a binary dependent variable) in both mares (normal vs. high risk-pregnancy) and foals (healthy vs. sick).

ALLO concentrations were expressed as median and interquartile range (IQR). A *p*-value of less than 0.05 was regarded as statistically significant. All statistical analyses were performed using SPSS software (Statistic version 25 IBM for Windows 11, SPSS Inc., Chicago, IL, USA).

## 3. Results

### 3.1. Clinical Data

The population included 37 animal pairs divided into healthy (group H; *n* = 15 pairs) and sick (group S; *n* = 22 pairs) foals. Table 1 presents the clinical data collected from mares and foals in the two groups.

In mares experiencing high-risk pregnancies, this condition was found to be linked to placentitis (7/22; 31.8%), placental edema (7/22; 31.8%), placental hypoplasia (4/22; 18.2%), twin pregnancy (2/22; 9.1%), and systemic illness (enteritis and laminitis, respectively; 2/22; 9.1%). Twelve out of 22 mares (54.5%) received treatment for the underlying condition.

Foals’ diagnoses included prematurity (6/22; 27.3%, PRE), dysmaturity (5/22; 22.7%, DYSM), hypoxic-ischemic encephalopathy (7/22; 31.8%, HIE), stillbirth (2/22; 9.1%), and congenital malformation (2/22; 9.1%). Nine out of 22 foals (41%) were born at the VTH and had a lower Apgar score than healthy ones (*p* = 0.048). Overall, sick foals were characterized by a lower weight at birth/admission (*p* < 0.001) and shorter mare’s gestation length (*p* < 0.001) than healthy foals.

### 3.2. ALLO Concentrations

Hair ALLO concentrations in foals and mares and the resulting ratios of H and S groups are shown in Table 2 and depicted in Figure 1. Foal ALLO concentrations and the foal/mare ALLO ratio were lower in group S than in group H (*p* < 0.001).

No significant differences in hair ALLO concentrations were observed across the entire population, including between males and females, between mares carrying male fetuses and those carrying female fetuses, between foals born at the VTH and those admitted post-birth, as well as between surviving and non-surviving foals. When considering all animals, no correlations were found between the foal and mare ALLO concentrations and mare’s age and parity, but moderate positive correlations were found between both foal ALLO concentrations and foal/mare ALLO ratio and mare’s gestation length (*p* = 0.003; r = 0.476 and *p* = 0.002; r = 0.487, respectively), and between foal/mare ALLO ratio and Apgar score at birth (*p* = 0.047; r = 0.410); eventually, a weak positive correlation emerged between foal ALLO concentrations and foal’s weight (*p* = 0.042; r = 0.336).

Subsequently, differences in ALLO concentrations were evaluated between healthy foals (group H) and sick foals classified according to diagnoses (S-PRE, *n* = 6; S-DYSM, *n* = 5; S-HIE, *n* = 7). Both foal ALLO concentrations and the foal/mare ALLO ratio were lower in each subgroup of sick foals than in group H (*p* < 0.001) but were not different between subgroups. Furthermore, the foal’s weight was lower in the S-PRE group than in the H group (*p* < 0.001) and S-HIE group (*p* = 0.025), and the mare’s gestation length was shorter in the S-PRE group than in the H group (*p* < 0.001), S-DYSM (*p* = 0.049) and S-HIE group (*p* = 0.027).

The logistic regression model revealed that ALLO concentrations had a low accuracy in predicting the clinical outcome of mares (normal vs. high-risk pregnancy; Nagelkerke’s R^2^ = 0.31, accuracy = 67.6, odds ratio = 0.995). Conversely, as reported in Table 3, a strong relationship (R^2^ = 0.75, odds ratio = 0.004) emerged between ALLO concentrations and foals’ clinical outcome (healthy vs. sick), with concentrations of the hormone predicting foals’ clinical outcome with high accuracy (86.8%). The inclusion of the variable (ALLO concentrations) was significant at *p* < 0.01.

## 4. Discussion

In the present study, the ALLO content in the hair of mares and their foals suffering from prenatal diseases was assesses for the first time. In group S, decreased foal ALLO concentrations and foal/mare ALLO ratio in the first 24 h after birth are suggested as possible retrospective biomarkers of feto-maternal well-being, although they are not diagnostic nor prognostic for the foals’ condition in the limited population of sick foals examined. From a clinical point of view, some interesting correlations with clinical parameters were found, which will be discussed.

The quantification of ALLO from equine hair has previously been validated using an ELISA kit to measure ALLO in both human and equine hair, but only in healthy pregnant and non-pregnant mares [17]. Later, ALLO concentrations were evaluated in the hair of mares longitudinally before and after parturition [18]. Comparing the results of the present study with those of the previous one [17], ALLO concentrations in mares (400.4 ± 152.0 pg/mg in group H and 416.2 ± 205.6 pg/mg in group S, respectively; data expressed as mean ± SD for comparison purposes only) were similar to those previously reported in mares during pregnancy (286.4 ± 141.3 pg/mg) or at parturition and much higher than those reported in non-pregnant mares (16.9 ± 5.5 pg/mg) and in a mixed population of healthy humans (7.3–79.1 pg/mg). ALLO, a neurosteroid and a key metabolite of progesterone, is synthesized in both the fetal brain and peripheral tissues, including the adrenal glands, ovaries, and testes [19]. During pregnancy, both in humans [20] and horses [8], ALLO plays a protective role against the effects of glucocorticoids released upon activation of the HPA axis. At the end of pregnancy, the reflex pathways responsible for initiating oxytocin secretion, which is crucial for parturition, become more sensitive. The premature stimulation of oxytocin release can lead to preterm birth. ALLO, however, is vital in preventing this early activation by inhibiting uterine muscle activity [20,21]. During late gestation, progesterone metabolites help maintain uterine quiescence in mares [22], and a notable decrease in circulating ALLO concentrations is observed beginning 5 days prior to parturition [9].

The first hypothesis, suggesting that ALLO concentrations found in the foal’s hair are influenced by those in the corresponding mare, has not been confirmed or, at least, remains under investigation. Unlike hair cortisol [6], no correlation was found between foal and mare hair ALLO concentrations under neither physiological nor pathological circumstances. This result could imply that prepartum ALLO deposition in fetal hair is influenced by multiple mechanisms in addition to variations in the maternal plasma ALLO concentrations, including those of the fetal central nervous system (CNS), adrenal glands, gonads, and allantochorion. It has been demonstrated that ALLO can cross the blood–brain barrier (BBB) in rats, and is thought to mediate its effects in the CNS via the GABA_A_ receptor [23]. The experimental infusion of ALLO in a healthy foal and subsequent induced changes in neurologic behavior and state of consciousness provided evidence that 5-alpha-reduced pregnanes are capable of crossing the BBB and exerting influence on the CNS [13]. While the exact origin of the exceptionally high pregnanolone concentrations in the serum of equine fetuses remains unclear, the experimental study results suggest that pregnanolone synthesized by the fetal gonads plays a significant role in the concentrations detected in fetal serum. This may also contribute notably to the placental synthesis of reduced 5-alpha pregnanes [24]. Pregnanolone concentrations in the fetal gonads and adrenal glands surpassed all other measured steroids in these tissues, being 100–150 times greater than the concentrations found in the allantochorion.

The second hypothesis, which proposed that foals categorized as sick at birth exhibit distinct hair ALLO concentrations when compared to healthy foals, was supported. During pregnancy, increasing ALLO concentrations function as an inhibitor of the HPA axis, which is believed to have stress-buffering effects on the mare and the fetus, thereby protecting them from the potentially detrimental effects of prolonged stress [20]. To date, only one study has explored the link between ALLO and anxiety symptoms in pregnant women during the second trimester. This study found an association between lower ALLO concentrations and elevated anxiety symptoms in response to stress [25]. Throughout pregnancy, the fetal brain has already developed the capability to synthesize ALLO independently [26], allowing the fetus to directly benefit from the increasing ALLO concentrations. These rising levels have been shown to support the growth of brain cells, nerve development, and myelination in the fetal brain [27]. Furthermore, ALLO exerts neuroprotective effects; in cases of pregnancy complications like intrauterine growth restriction or preterm birth, ALLO concentrations rise and stimulate rapid cell proliferation, which helps reduce brain damage and safeguards immature fetal brain cells [28,29]. In late pregnancy, ALLO may prevent preterm birth by inhibiting oxytocin through the stimulation of the opioid system and regulating the expression of the oxytocin gene in the mother [30]. These findings from human medicine suggest that the increase in ALLO concentrations during pregnancy plays an adaptive role in supporting fetal development. It cannot be excluded that the decrease in ALLO deposition in the hair of sick foals reported in this study may help protect them from the harmful effects of chronic fetal stress and hypoxia. On the other hand, the inefficiency of the feto-placental unit that characterized the mares with high-risk pregnancies selected in the present study is a further element to be considered when assessing the reduced concentrations of ALLO deposited in the hair of the respective foals, altered vascular permeability (as in the course of placental edema and placentitis) being a possible limiting mechanism in the trans-placental diffusion of steroid hormones. When ALLO concentrations were compared among sick foals according to different diagnoses, no differences were found among foals affected by prematurity, dysmaturity, or hypoxic-ischemic encephalopathy. This result is probably due to the small sample size and the fact that hormone concentrations cannot be considered a clinical diagnostic tool. Because these reported conditions often share a common feature of altered feto-placental “communication”, the biomarker is unlikely to be able to discriminate among them. Although premature foals had a lower weight at birth/admission and a shorter gestation length than the other foals, ALLO concentrations in the population of sick foals would not be expected to be influenced by the presence of the few premature foals, as the same differences between sick and healthy foals were found in all diagnostic categories.

Finally, the ALLO concentration showed strong potential in predicting the foals’ clinical outcome; in contrast, it was not able to differentiate mares with normal pregnancy from those with high-risk pregnancy, considering the limited size of the population, the variability within the groups, and the medical treatments to which mares with high-risk pregnancies were subjected. The inability to differentiate the type of pregnancy in mares was given by the similar range of ALLO values recorded, as discussed above, whereas in foals, only a few ALLO records overlapped between the H and S groups.

From a clinical perspective, moderate positive correlations were found between foal ALLO concentrations and foal/mare ALLO ratio and mare’s gestation length and between foal ALLO concentrations and foal’s weight. Limited data are available in the human literature, primarily focusing on differences in circulating ALLO based on sex or age. In general, ALLO concentrations are higher during the fertile years and pregnancy, and decline with age in men, but remain stable in women [31,32,33]. In the equine species, it is possible to hypothesize that the deposition of ALLO in the hair of the fetus follows the normal progression of pregnancy in terms of days and the physiological weight gain of the fetus itself until final maturation is reached. A moderate positive correlation was also found between the foal/mare ALLO ratio and the Apgar score [16], considering the neuroprotective effects of ALLO in the CNS. The Apgar score offers a semi-quantitative evaluation of the severity of signs resulting from peripartum asphyxia, using a 0–10 scoring system ranging from severe (<6) and mild (6–8) asphyxia to normal foals (8–10). Although the foal/mare ALLO ratio did not appear to be diagnostic for foals specifically affected by hypoxic-ischemic encephalopathy compared to premature or dysmature ones, it cannot be ruled out that further studies involving larger populations may enhance the diagnostic sensitivity of this biomarker in foals that suffered from hypoxia during intra-uterine life. Nevertheless, any speculative conclusions about correlations with clinical parameters should be carefully screened in future studies, as none of the parameters showed a strong correlation. Overall, further research is encouraged to generalize the findings of the present study, potentially involving a larger sample size in order to achieve greater statistical power and improve the diagnostic sensitivity of ALLO.

## 5. Conclusions

Since the prenatal activity of the HPA axis is crucial for the ultimate maturation of the fetus and its adaptation to extra-uterine life, and that ALLO attenuates the HPA axis response in late pregnancy, the lower ALLO concentration in sick foals observed in the present study deserves further attention as a potential new biomarker of prenatal disease, considering that its concentration in hair from mares did not carry retrospective information on the type of pregnancy.

## Figures and Tables

**Figure 1 animals-15-00768-f001:**
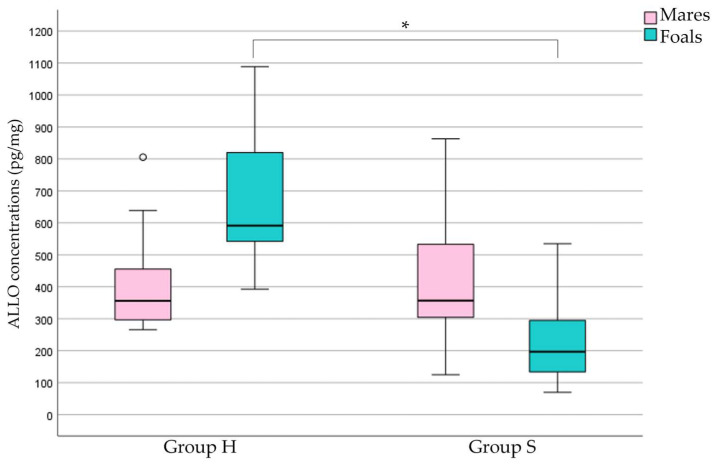
ALLO concentration comparison between mares and foals in the two groups. * A significant difference was observed between healthy and sick foals (heathy–H and sick–S; *p* < 0.001) with Mann–Whitney test. Circle placed past the whiskers indicates outlier.

**Table 1 animals-15-00768-t001:** Clinical data for mares and foals in the two groups were collected. The data are presented as mean ± standard deviation (min–max). STB = Standardbred; SDB = Saddlebred; QH = Quarter Horse; N = normal pregnancy; HR = high-risk pregnancy; Sv = survived to hospital discharge; NSv = not survived. * Superscript asterisk in the column indicates a significant difference between groups with Mann–Whitney test.

Mares	Breed	Age (Years)	Parity (n)	Type of Pregnancy (N/HR)	Gestation Length (Days)
Group H (*n* = 15)	STB *n* = 13 SDB *n* = 2	11 ± 6 (5–24)	5 ± 4 (1–14)	N *n* = 15	341 ± 10 (326–359)
Group S (*n* = 22)	STB *n* = 14 SDB *n* = 6 QH *n* = 2	12 ± 4 (5–20)	4 ± 4 (1–11)	HR *n* = 22	324 ± 16 (281–350) *
**Foals**	**Sex**	**Weight** (Kg)	**Apgar score**	**Age at** **Admission** (Hours)	**Outcome**
Group H (*n* = 15)	Male *n* = 6 Female *n* = 9	49 ± 6 (38–60)	9 ± 1 (8–10) *n* = 14	0	Sv
Group S (*n* = 22)	Male *n* = 12 Female *n* = 10	40 ± 8 (23–55) *	8 ± 1 (6–10) *n* = 10 *	5 ± 7 (0–21)	Sv *n* = 15 NSv *n* = 7

**Table 2 animals-15-00768-t002:** The results of ALLO concentrations in the two groups are presented. Data are expressed as the median and interquartile range (IQR). * Superscript asterisk in the row denotes a statistically significant difference (*p* < 0.001) between the groups, as determined by the Mann–Whitney test.

	Group H (*n* = 15 Pairs; 30 Animals)	Group S (n = 22 Pairs; 44 Animals)
Mare ALLO (pg/mg)	356.0 (158.4)	356.7 (217.8)
Foal ALLO (pg/mg)	591.1 (215.9)	197.0 (154.3) *
Foal/Mare ALLO ratio	1.6 (1.2)	0.6 (0.5) *

**Table 3 animals-15-00768-t003:** Logistic regression model used to study the predictive potential of ALLO concentrations in relation to clinical outcome in foals (healthy vs. sick).

Likelihood Logarithm	Cox and Snel R-Square	Nagelkerke R-Square	Overall Accuracy (%)
20.795	0.557	0.749	86.8

## Data Availability

The original contributions presented in this study are included in the article. Further inquiries can be directed to the corresponding author.

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
