# Peer review of "Hair Allopregnanolone in Mares and Foals as a Retrospective Biomarker of Predicting Feto-Maternal Well-Being"

_animals, 2025, doi:10.3390/ani15060768_

Round 1

Reviewer 1 Report

Comments and Suggestions for Authors

Dear editors,

In this paper, the authors evaluated the possible role of allopregnanolone (ALLO) as a biomarker predicting feto-maternal well-being by analyzing hair of 37 mares with normal or high-risk pregnancies and their respective healthy and sick foals. They highlighted the lower ALLO concentrations in sick foals potential, suggesting a role for this steroid as a new biomarker of prenatal disease. Based on this, this research sheds light on the hypothalamus-pituitary-adrenal (HPA) axis maternal-fetal interplay, offering new insights into the role of pregnane functions during pregnancy. The topic is very interesting and current. The manuscript is well structured. The references are appropriate. However, the English should be revised in some sections, as some sentences are overly complex and should be simplified.

Abstract and summary: they are very well structured recapping the information contained in the main text without repetitions.

Key-words: they are pertinent and consistent with the topic.

Introduction: This section properly shows the state-of-the-art resuming the knowledge about this topic. If it has been explained before, repeating the meaning of the acronym ALLO is not required. The aim of the study and the hypothesis are clearly expressed.

Materials and Methods: They are well structured and properly describes the sample sizes, the procedures and statistical tests. However, I have some suggestion: it would be proper to move the text about clinical data (lines 145-148 and 200-202) to the “Population” paragraph, as this information are referring to the enrolled animals in the study. Also, Table 1 should be moved to the same paragraph (these data are not results). Could you provide some information about the enrolled mares? I.e. general information about the type of diet. It would also be suitable to add the intra- and inter-assay coefficients of variation (CVs) for the used test.

Results: They are well presented and joined with very detailed images and tables, adding information to the main text. However, as previously mentioned, this paragraph should be fixed, moving part of this to the “Materials and Methods” paragraph.

Discussion: this section is logically written with compelling interpretations. However, the limits of this study should be discussed thoroughly. Line 291: please, specify which species reference 25 is referring to. Lines 317-318: please, indicate which species this conclusion refers to, based on the references cited in the preceding sentences.

Conclusions: It properly hypothesizes the future applications of this discovery.

Overall the paper is very captivating and, since it adds useful information in the field of reproductive endocrinology, it deserves to be published after minor revisions.

Reviewer 2 Report

Comments and Suggestions for Authors

Line 58: 270 to 335 days is not the last third of pregnancy, but the last two months of pregnancy. The last third of pregnancy is from 220 days of pregnancy to foaling. Since hair starts to grow at 270, it is not correct to say the last third of pregnancy. Please correct. 

Line 103: Iformation should not be blinded for review in this journal.

Line 113: Please  give reference for what an increased CTUP was considered.

Experimental design: There is no information on when the hair sample was obtained from mares which did not foal at the VTH. Please provide clear information on the timing of the hair sample in mares. The results indicate no difference in ALLO concentration between H and S mares. These should be discussed. Furthermore, there is scarce information on the inclusion criteria and source of mares for the study. Was it a convenience criteria or randomnly chosen?

Conclusions are speculative, as the correlations were moderate to weak. Furthermore, the ALLO test at birth does not provide any added information than what is already available from the clinical parameters (high pregnancy risk). It was not possible to associate retrospectively a lower ALLO concentrations in mares from high risk pregnancies, which would help to predict in advance a compromised foal.

Statistics: The sample size of 37, with 15 couples in the H group is too low to perform a multivariate regression model. Please explain how did you calculate the power sample size of your study?

The OR for the binary logistic regression model is missing.

correlation coeficients (r) < 0.4 represent a weak level of assocaition, while the authors report it as "moderate" (Line 230). 

Reviewer 3 Report

Comments and Suggestions for Authors

General comments:

This study evaluated the foal and mare hair allopregnanolone concentration during the first 24h after parturition. The relationships between several physiological or clinical variables and allopregnanolone levels were assessed. Two hypotheses are made and supported or rejected by the results. This is a relevant subject, and the study shows scientific soundness. Limitations of the study were reported by the authors. The introduction contextualizes and justifies the aims and both hypotheses. A full description was given in M&M. Due to the different sources of the sampled population (mares and foals) a deep description was made. Maybe a figure summarizing this point can simplify the readability of this part (2.1 Population). The statistical analysis is appropriate, and the results fully described. The findings are very well discussed, supported by appropriate literature. The discussion was not limited to comparisons between studied and authors deeply approach potential justifications of the findings, including the limitations of the study. The conclusion is supported by the results. No major issues were found.

Specific comments:

L258: Decreased ALLO in group S.

L270: Relatively similar or to a similar degree.

L270-272: The data was only from [19].

Round 2

Reviewer 2 Report

Comments and Suggestions for Authors

The statistical analyses are not correct yet. A inary logistic regression is reported with P Value and odds ratio, not with a correlation coefficient. 

COnclusions are still speculative as the ALLO concentration does not add any clinical information compared to the inforrmation allready obtained by examining the foal.
